# Regulation of Mitochondrial Metabolism by Hepatitis B Virus

**DOI:** 10.3390/v15122359

**Published:** 2023-11-30

**Authors:** Yumei Li, Jing-hsiung James Ou

**Affiliations:** Department of Molecular Microbiology and Immunology, University of Southern California Keck School of Medicine, Los Angeles, CA 90033, USA; lymgz22@gmail.com

**Keywords:** hepatitis B virus, mitochondrial metabolism, oxidative phosphorylation, innate immune response, mitophagy, CD8^+^ T cells, hepatic macrophages

## Abstract

Mitochondria play important roles in the synthesis of ATP, the production of reactive oxygen species, and the regulation of innate immune response and apoptosis. Many viruses perturb mitochondrial activities to promote their replication and cause cell damage. Hepatitis B virus (HBV) is a hepatotropic virus that can cause severe liver diseases, including cirrhosis and hepatocellular carcinoma (HCC). This virus can also alter mitochondrial functions and metabolism to promote its replication and persistence. In this report, we summarize recent research progress on the interaction between HBV and mitochondrial metabolism, as well as the effect this interaction has on HBV replication and persistence.

## 1. Introduction

Mitochondria are cellular organelles critical for energy production, calcium homeostasis, and cell signaling. They also play crucial roles in antiviral responses, and many viruses are known to modulate mitochondrial functions to enhance their replication and survival [1,2]. Mitochondria are involved in multiple metabolic pathways, including oxidative phosphorylation (OXPHOS); the tricarboxylic acid cycle (TCA cycle, also known as the Krebs cycle); β-oxidation for the conversion of fatty acid into acetyl-CoA for its entry into the TCA cycle; the production of reactive oxygen species (ROS); and the regulation of apoptosis (Figure 1). OXPHOS involves the transfer of electrons through a series of protein complexes located in the inner mitochondrial membrane; this results in the synthesis of ATP, which is the primary source of cellular energy (Figure 1). The TCA cycle is a key part of aerobic respiration. It generates NADH and FADH_2_ to fuel electron transport chains for ATP production. Viruses can perturb mitochondrial metabolism to cause damage to cells and to promote their replication. For example, Zika virus (ZIKV) can alter NAD^+^-related metabolic pathways, including the downregulation of the TCA cycle and OXPHOS in neurons and neuroblasts, which can cause microcephaly in mouse brains [3]. Human immunodeficiency virus type-1 (HIV-1) has also been shown to alter the metabolism of CD4^+^ T cells by elevating OXPHOS to promote its replication [4]. Hepatitis C virus (HCV) core protein can also localize to the outer mitochondrial membrane, resulting in the inhibition of mitochondrial electron transport and an increase in the production of ROS [5]. HCV has also been shown to induce dynamin-related protein 1 (Drp1)-mediated mitochondrial fission and mitophagy (i.e., the selective removal of mitochondria by autophagy) to support its persistence and replication, and the suppression of this process reduced glycolysis, ATP production, and viral release [6]. HCV infection also alters the lipid metabolism of its host cells to promote its replication (for a review, see [7]). This alteration may be caused by a variety of factors, including the induction of a metabolic shift toward glycolysis, which can generate acetyl-CoA, an intermediate in triglyceride synthesis. This leads to the formation of lipid droplets as well as a reduction in fatty acid β-oxidation [8]. There are also other viruses that can alter the mitochondrial metabolism of cells. In this review, however, we will focus on hepatitis B virus (HBV) and summarize recent advances in our understanding of the relationship between HBV and mitochondrial metabolism. 

## 2. HBV and Its Life Cycle

HBV is one of the most important human pathogens. It can cause severe liver diseases, including acute and chronic hepatitis, cirrhosis, and hepatocellular carcinoma (HCC). HBV chronically infects approximately 300 million people in the world and causes an estimated 820,000 deaths annually [9]. Most chronic HBV patients acquire the virus from their mothers early in life. In contrast to this mother-to-child transmission, which is also known as vertical transmission, horizontal transmission between two unrelated individuals (e.g., through the sharing of drug injection needles or via sexual activities) usually leads to self-limited acute infection [10]. HBV does not cause lytic infection and, as such, there are many asymptomatic HBV carriers. HBV-induced liver injury may involve the appearance of HBV variants and is often associated with the activation of the host CD8^+^ T cell response, which is impaired during chronic HBV infection (for a review, see [11]). 

HBV is a small DNA virus that belongs to the hepadnavirus family. It has a very narrow host range and infects only humans and a few primate species [12]. In 1970, Dane and colleagues visualized the virus particles using an electron microscope, revealing the presence of three types of particles in the serum of infected patients: spherical particles with a diameter of 42 nm, spherical particles with a diameter of 22 nm, and filamentous particles with a diameter of 22 nm [13,14]. The 42 nm spherical particles, which have since been called Dane particles, are mature virions consisting of a lipid envelope with three co-carboxy-terminal viral envelope proteins called large (L), middle (M), and small (S) surface antigens (HBsAg). The Dane particle contains a core that displays the core antigenic determinant known as the core antigen (HBcAg). The major protein constituent of this core particle is the core protein, which forms a protein shell that surrounds the viral DNA genome. Studies in recent years have indicated that more than 90% of 42 nm particles do not contain the HBV DNA genome [11,15]. The HBV genome is a circular and partially double-stranded DNA molecule with a length of approximately 3.2 kilobases. It encodes four genes: C, P, S, and X. The C gene codes for the core protein and a related protein termed the precore protein; the P gene codes for the viral DNA polymerase, which is also a reverse transcriptase; the S gene codes for the three HBsAg proteins; and the X gene codes for the X protein (HBx). The precore protein is the precursor of the e antigen (HBeAg) found in the serum of HBV patients. In contrast to Dane particles, which contain a core, 22 nm particles are empty envelope particles. In addition to Dane particles and 22 nm particles, naked core particles are also released from HBV-infected cells [16,17]. 

HBV is a hepatotropic virus and enters hepatocytes via an interaction with its receptor (i.e., sodium taurocholate cotransporting polypeptid (NTCP)) on the cell surface [18]. Once inside the cell, the core particle delivers the relaxed circular genomic DNA (rcDNA) into the nucleus, where it is converted to a covalently closed circular DNA (cccDNA). This cccDNA serves as the transcription template for all viral mRNAs. The core protein mRNA is larger than the genome and is known as the pregenomic RNA (pgRNA). It also codes for viral DNA polymerase. Once the core protein is synthesized, it packages the pgRNA to form the core particle, in which the pgRNA is reverse transcribed and converted to rcDNA by the viral DNA polymerase that is also packaged [11]. The core particles containing the rcDNA may interact with HBsAg in intracellular membranes to form mature viral particles for release from infected hepatocytes. Alternatively, they may transport the rcDNA back into the nucleus for repair and amplification of the cccDNA pool. HBV DNA can also integrate into host chromosomes. Although the integrated HBV DNA may also direct the synthesis of viral RNA transcripts, this integration is not an essential step in the viral life cycle [19,20]. 

## 3. HBV and Host Metabolism

HBV infection can cause metabolic changes in patients. In an untargeted metabolomic analysis of serum samples of 199 HBsAg-positive patients with active (i.e., serum HBV DNA > 100 IU/mL) and inactive (i.e., serum HBV DNA < 100 IU/mL) HBV replication and different states of liver diseases including chronic hepatitis, cirrhosis and HCC, significant alterations of metabolites, including a decrease in amino acids and an increase in phosphatidylcholines, were observed in patients with active HBV replication [21]. In a separate targeted metabolomic analysis of serum samples, significant metabolic alterations, including increases in free fatty acids, acylcarnitines, and plasmalogens, and decreases in triglycerides, phospholipids, and sphingomyelins, in the immune tolerance phase of HBV infection (i.e., HBsAg and HBeAg-positive with a mean serum HBV DNA copy number of 8.82 logs and a serum alanine aminotransferase [ALT] level < 40 U/L) were also found [22]. An analysis of serum ATP levels also revealed significantly lower ATP levels in chronic HBV patients than in healthy controls, likely due to impaired mitochondrial functions in HBV patients [23]. These findings indicated that HBV may affect the metabolic pathways in patients.

### 3.1. Effect of HBV on the Metabolism of Hepatocytes

HBV can cause changes of metabolism in hepatocytes [24]. When an adenovirus vector was used to deliver an overlength HBV genome into rat hepatocytes, which served as a surrogate model for HBV-infected human hepatocytes, it was found that HBV could alter the metabolome of hepatocytes. Further transcriptomic analysis by the authors revealed the alteration of metabolic pathways that included long-chain fatty acid metabolism, glycolysis, and glycogen metabolism [25]. The glycerol-3-phosphate (G3P) shuttle is a pathway that translocates electrons produced during glycolysis across the inner membrane of mitochondria for OXPHOS. It is mediated by the combined activities of cytosolic G3P dehydrogenase (cGPDH or GPD1) and mitochondrial FAD-dependent G3P dehydrogenase (mGPDH or GPD2). In a study to examine the interaction between HBV and the G3P shuttle, it was found that GPD2, but not GPD1, suppressed HBV replication by recruiting the E3 ubiquitin ligase TRIM 28 to form a complex with HBx, a regulatory protein of HBV, resulting in the proteasomal degradation of HBx [26]. This reduction of HBx by GPD2 does not require the enzymatic activities of GPD2 and is likely a host defense mechanism against HBV. This may also be the reason why HBV reduces the level of GPD2 in liver tissues [26]. The reduction of GPD2 by HBV likely affects the G3P pathway, which could explain the reduced glycerophospholipid and increased plasmalogen species found in the serum of chronic HBV patients [22].

#### 3.1.1. Role of HBx on Mitochondria Metabolism in Hepatocytes

The HBx protein of HBV is a regulatory protein that has multiple functions. It can activate transcription factors, regulate signaling pathways, promote hepatocarcinogenesis, and, more importantly, enhance HBV gene expression (for detailed reviews, see [11,27]). It can be localized to mitochondria via multiple regions of its sequence [28,29,30]. It can also interact with the voltage-dependent anion channel (VDAC) [31], a channel that spans across the mitochondrial outer membrane, and cytochrome c oxidase subunit III (COXIII), a protein that spans multiple times across the inner mitochondrial membrane and a subunit of the respiratory complex IV [32,33]. HBx can activate NF-κB to prevent the depolarization of mitochondrial membranes. However, when NF-κB is inhibited, HBx will induce the depolarization of mitochondrial membranes [30]. HBx can also induce the expression of cyclooxygenase-2 (COX-2) through the induction of ROS to promote cell growth [34]. In contrast, MARCH5, another E3 ubiquitin ligase that is localized to the outer membrane of mitochondria, can also interact via its N-terminal RING domain with HBx to mediate the degradation of HBx [35] (Figure 1). This leads to the suppression of HBx-induced ROS production, mitophagy, and COX-2 gene expression. 

HBx has also been shown to bind to Orai1, a store-operated calcium entry component on the plasma membrane, to increase intracellular calcium levels via an influx of extracellular calcium [36]. This influx leads to an increase in calcium uptake by VDAC into mitochondria and the activation of calcium signaling pathways [37] (Figure 2). Thus, HBx exerts multiple effects on mitochondria to cause their dysfunction [38].

#### 3.1.2. Role of Mitochondria in Innate Immune Responses against HBV

Cellular metabolism is often linked to cellular innate immune responses. Retinoic acid-inducible gene I (RIG-I)-like receptors (RLRs), which include RIG-I and MDA5, are pattern recognition receptors (PRRs) that are activated by pathogen-associated molecular patterns (PAMPs) present on RNA molecules. Upon their activation, they will bind to a signaling adaptor named mitochondrial antiviral signaling (MAVS) on mitochondrial membranes to activate the downstream signaling for the induction of type I interferons (IFNs) and pro-inflammatory cytokines. Recently, it was shown that HBV promoted glycolysis to suppress RIG-I-mediated IFN induction [39]. In this elegant study, it was shown that primary human hepatocytes (PHH) infected by HBV had elevated levels of metabolic intermediates, including pyruvate, lactate, and other metabolic intermediates of the TCA cycle. The increase in lactate was due to the activation of hexokinase (HK) and lactate dehydrogenase A (LDHA) by HBV. Next, Lactate would then bind to MAVS to prevent it from interacting with RIG-I on mitochondrial membranes (Figure 2). In addition, HBV also sequestered MAVS by promoting the formation of a ternary complex of MAVS, HK2, and VDAC on mitochondrial membranes (Figure 2). These activities of HBV on MAVS disrupted its downstream signaling and the subsequent induction of IFNs and enhanced HBV replication [39]. These studies indicated that HBV could modulate cellular metabolism to evade innate immune response to benefit its own replication. 

HBV also stimulated the expression of interleukin-6 (IL-6), a cytokine that activates STAT3 to promote hepatocellular proliferation, in liver tumor cell lines and liver tissues of HBV patients [40]. This stimulation of IL-6 expression was due to the induction of ROS in mitochondria by HBV, as it could be abolished if cells were treated with rotenone, which blocked ROS production in mitochondria. ROS upregulated the expression of the transcription factor Nrf2, which stimulated the expression of IL-6. Suppressor of cytokine signaling 3 (SOCS3) is a negative feedback regulator of the IL-6/STAT3 signaling pathway. Interestingly, the expression of SOCS3 was also blocked by ROS, which induced the expression of the transcription factor Snail. Snail binds to the E-boxes of the SOCS3 promoter and mediates its epigenetic silencing in association with DNA methyltransferase 1 (DNMT1) and histone deacetylase 1 (HDAC1). The silencing of SOCS3 led to sustained activation of the IL-6/STAT3 pathway, which likely contributed to HBV-induced hepatocarcinogenesis [40] (Figure 2).

#### 3.1.3. Integration of HBV DNA into Mitochondrial DNA

HBV DNA can integrate into host chromosomes, and this integration may activate or disrupt cellular genes to promote hepatocarcinogenesis (for a review, see [41]). Interestingly, in a recent study it was shown that HBV DNA could also integrate into mitochondrial DNA in OXPHOS genes and the *D-loop* region, which resides in the main noncoding region of mitochondrial DNA [42]. The HBV DNA breakpoints were enriched in the preS/S region of the viral genome when it was integrated in the *D-loop*, and in the X gene (including its upstream enhancer I/X promoter and the downstream precore sequence) when it was integrated in the mitochondrial protein coding sequences [42]. The integration of HBV DNA encompassing the X and precore sequence in the mitochondria-encoded cytochrome c oxidase III (MT-CO3) gene was also reported in another study [43]. In this study, both HBV DNA and the host DNA at the integration junctions were found to be hypermethylated. How HBV DNA integrates into mitochondrial genome is unclear. However, as HBV RNA sequences were also detected in mitochondria, it was suggested that the HBV DNA integration might be mediated by HBV RNAs [42]. The biological significance of this integration is also unclear, although it suggests the possibility that HBV may also regulate mitochondrial metabolism via the disruption of the mitochondrial genome.

#### 3.1.4. HBV and Mitophagy

HBV may also affect mitochondrial metabolism by inducing mitophagy, the selective removal of mitochondria by autophagy. Kim et al. reported that HBV or its HBx protein could induce the perinuclear clustering of mitochondria and the localization of Drp1 to mitochondria by stimulating its phosphorylation at serine-616 [44]. The localization of Drp1 to mitochondria can trigger mitochondrial fission [45], resulting in the segregation of depolarized mitochondria that are targets for removal by mitophagy [46]. Note that the induction of mitochondrial clustering is not a unique property of HBV, as other viruses such as rubella virus [47] and respiratory syncytial virus [48] have also been shown to exert such an effect on mitochondria. Kim et al. found that, in addition to inducing mitochondrial fragmentation, HBV or its HBx protein could also induce the expression of PINK1, Parkin, and LC3B, which are protein factors critical for mitophagy. PINK1 is a serine/threonine kinase and a labile protein. However, it is stabilized on depolarized mitochondria and its accumulation on mitochondria can recruit and activate Parkin, an E3 ubiquitin ligase, to promote the ubiquitination of mitochondrial outer membrane proteins to trigger mitophagy. Parkin activated by HBV could also cause the degradation of Mitofusin 2, a mediator of mitochondrial fusion, and initiate mitophagy [44]. The induction of mitophagy by HBx could attenuate mitochondria-related apoptosis [44] and also trigger a metabolic shift toward glycolysis [49]. As mentioned above, the activity of HBx in the induction of mitophagy could be antagonized by MARCH, an E3 ubiquitin ligase, which binds to HBx on mitochondrial membranes to induce its degradation [35]. Interestingly, although HBx induced mitophagy, the large HBsAg (L-HBsAg) was also recently shown to inhibit sorafenib-induced mitophagy via the WNT7B/CTNNB1 signaling pathway; therefore, it may play a role in chemoresistance to this HCC drug [50]. 

#### 3.1.5. HBV preS2 Mutant and Mitochondrial Metabolism

The HBV middle HBsAg (M-HBsAg) is not essential for HBV replication [51,52]. HBV mutants that contain in-frame deletions at the N-terminus of the preS2 coding sequence involving the M-HBsAg ATG codon are often isolated from chronic HBV patients during natural infection. These preS2 mutants cannot express the M-HBsAg and express a L-HBsAg with an internal deletion that contains T-cell and B-cell epitopes [53]. Due to the loss of T-cell and B-cell epitopes in the L-HBsAg, HBV preS2 mutants likely represent immune-escape mutants and are associated with more severe liver diseases, including HCC. A recent study indicated that preS2 mutants could cause the endoplasmic reticulum (ER) stress, reduce the mitochondrial membrane potential and ATP production, and cause calcium overload, which may be associated with its increased pathogenicity [54]. 

### 3.2. Effect of HBV on the Metabolism of CD8 Cells

CD8^+^ cytotoxic T lymphocytes (CTLs) play a critical role in the removal of HBV from infected individuals. However, in patients with chronic HBV infection, HBV-specific CD8^+^ T cells are functionally exhausted with dysfunctional mitochondria [55,56]. Indeed, in our recent study using mice as a model, we found that HBV-specific CD8^+^ T cells expressed a high level of programmed cell death protein 1 (PD-1), an immune checkpoint protein, in mice with persistent HBV replication, and that the functionalities of these CD8^+^ T cells were impaired [57]. Fisicaro et al. compared the transcriptomic profiles of HBV-specific CD8^+^ T cells isolated from patients with acute and chronic diseases with those of patients with resolved HBV infection [55]. They also used influenza-specific CD8^+^ T cells isolated from healthy individuals as a control for comparison. They found that exhausted HBV-specific CD8^+^ T cells had extensive mitochondrial alterations, and that the antiviral functions of those CD8^+^ T cells could be improved by antioxidants that targeted mitochondria, suggesting a role for ROS in T cell exhaustion [55]. This result was confirmed by Acerbi et al. [56]. Their findings raised the possibility of targeting mitochondria to treat chronic HBV patients. 

The increased production of ROS by dysfunctional mitochondria can lead to DNA damage. In a recent report, it was found that exhausted HBV-specific CD8^+^ T cells had increased DNA damage and dysfunctional repair [58]. This DNA damage was associated with an increased expression of CD38, also known as cyclic ADP ribose hydrolase, a major NAD consumer, as well as the persistent activation of poly-ADP-ribose polymerase (PARP), which led to enhanced consumption of NAD and its depletion. The loss of NAD impaired poly-ADP ribosylation (i.e., PARylation), which is important for DNA damage repair [59], and amplified cellular dysfunction of exhausted HBV-specific CD8^+^ T cells (Figure 3A). In further studies, it was shown that the replenishment of NAD could restore CD8^+^ T cell functions [58]. Interestingly, Schmidt et al. showed that the inhibition of acyl-CoA:cholesterol acyltransferase (ACAT), which catalyzes the esterification of cholesterol for its storage in lipid droplets in the cytoplasm, could enhance OXHPOS and the production of ATP, and thereby rescue dysfunctional HBV-specific CD8^+^ T cells ex vivo [60]. Thus, the modulation of mitochondrial metabolism offers a new avenue for the treatment of chronic HBV patients. How HBV affects the mitochondrial metabolism of CD8^+^ T cells remains unclear, as HBV does not infect these cells. It is conceivable that this may involve the effects of other immune cells and/or cytokines. Further studies will be required to address this question.

### 3.3. Effect of HBV on the Metabolism of Macrophages

Macrophages are phenotypically plastic and can undergo M1 pro-inflammatory polarization or M2 anti-inflammatory polarization with different metabolic profiles [61,62]. M1 macrophages, which are characterized by their ability to produce ROS and pro-inflammatory cytokines such as IL-1β and TNF-α, frequently display high glycolytic activities and low OXPHOS activities. In contrast, M2 macrophages, which are characterized by their expression of CD163 (a member of the scavenger receptor superfamily), mannose receptor C type 1 (MRC1), and IL-10 (an anti-inflammatory cytokine), display low glycolytic activities and high OXPHOS activities [63]. However, recent studies have indicated that the dichotomous separation of M1 and M2 metabolic macrophage phenotypes might be over-simplified [64]. In HBV infection, macrophages play a crucial role in both viral clearance and viral persistence. By using HBV transgenic mice as a model, we previously demonstrated that genotypically HBV-negative mice born to hemizygous HBV transgenic dams were immunotolerant to HBV. Hence, HBV could establish persistent replication in these mice, if HBV genomic DNA was introduced into the liver of these mice by hydrodynamic injection [57]. In contrast, HBV was not able to establish persistence if its genomic DNA was injected into mice that were born to HBV-negative dams. These results indicated that maternal HBV antigens affected an offspring’s immunity and provided an explanation as to why the mother-to-child transmission of HBV frequently leads to lifelong chronic HBV infection in children. Conversely, horizontal transmission between two unrelated individuals, such as via the sharing of drug injection needles or sexual activity, frequently leads to self-limited acute HBV infection [10]. Our further studies indicated that mouse Kupffer cells, the resident macrophages of the liver, underwent M1-like polarization if they had not been exposed to maternal HBV antigens but underwent M2-like polarization if they had been exposed [10,65]. These findings were consistent with a previous report, which indicated that chronic HBV infection in a humanized mouse model grafted with human hepatocytes and hematopoietic stem cells was associated with a high level of infiltrating M2-like macrophages [66]. They were also consistent with the clinical observation that patients with lower HBV DNA and higher ALT levels have more CD16^+^ monocytes and/or macrophages in the peripheral blood and liver, indicating an immune-activated state. CD16^+^ is a marker of M1-like macrophages [67]. 

HBV can also affect the metabolism of macrophages and hence alter their functions, leading to impaired immune responses and increased susceptibility to infection [68,69]. The disruption of mitochondrial metabolism in macrophages can lead to decreased energy production and increased oxidative stress [24]. This can have a range of negative effects on macrophage functions, including impaired phagocytosis, decreased cytokine production, and altered polarization (i.e., a process by which macrophages differentiate into different functional states). By conducting RNA-seq to examine the effect of HBV on macrophages derived from THP-1 cells, a human monocytic cell line, we found that HBV induced the expression of mitochondrial enzymes that are involved in OXPHOS [65]. Further pathway analysis and the measurement of the oxygen consumption rate (OCR) and extracellular acidification rate (ECAR) (which reflect OXPHOS and glycolysis activities, respectively) confirmed that HBV could induce OXPHOS and inhibit glycolysis in THP-1 macrophages (Figure 3B). The same results were obtained, when Kupffer cells isolated from naïve mice that had been hydrodynamically injected with the HBV genomic DNA. These Kupffer cells displayed M1-like phenotypes as mentioned above. The finding that HBV could induce OXPHOS in M1-like macrophages is rather surprising, as previous studies indicated that M1-like macrophages often had enhanced glycolytic activities and low OXPHOS activities [63]. Our finding indicates that the metabolic reprogramming in macrophages is complex, and that different external stimuli can lead to different metabolic outcomes. Since the treatment of macrophages with dimethyl malonate (DMM), which inhibits succinate oxidation to inhibit OXPHOS, reduced the expression of IL-1β, this induction of OXPHOS by HBV could reduce the production] of IL-1β by macrophages [65]. This finding is important, as IL-1β by binding to its receptor on hepatocytes could suppress HBV gene expression and HBV replication [65] (Figure 3B). Thus, this induction of OXPHOS in macrophages is used by HBV to attenuate this antiviral response. As HBV-producing cell lines co-cultured with macrophages using Transwell can induce the OXPHOS of macrophages [65], HBV likely exerts its effect on macrophages via antigens released from cells. 

## 4. Conclusions

Mitochondria play important roles in ATP synthesis, innate immune response, and apoptosis. Studies in recent years have revealed that HBV uses multiple pathways to affect mitochondrial metabolism in both hepatocytes and immune cells. HBV can alter the physiology and metabolism of mitochondria to induce calcium signaling and disrupt host innate immune responses. It can also promote mitochondrial fission and mitophagy and suppress the antiviral activities of immune cells. These effects of HBV on mitochondria promote HBV replication, persistence, and pathogenesis. As such, metabolic pathways have also emerged as targets for the development of drugs to treat chronic HBV patients.

## Figures and Tables

**Figure 1 viruses-15-02359-f001:**
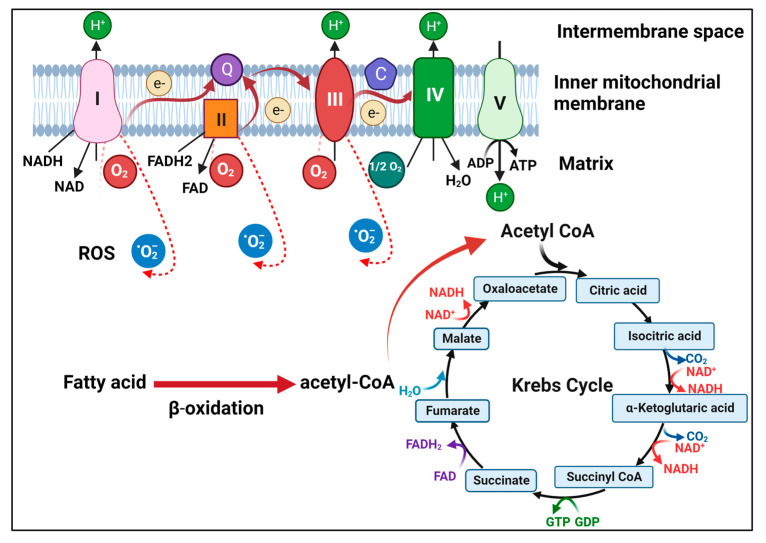
Illustration of mitochondrial metabolism. Fatty acids undergo β-oxidation to generate acetyl-CoA, which then drives the TCA cycle in the matrix of mitochondria. NADH and FADH_2_ generated by the TCA cycle donate electrons to complex I and complex II, respectively, on the inner mitochondrial membrane. The electrons are then transferred through ubiquinone (Q) to complex III and then via cytochrome C (C) to complex IV. The electrons are then transferred to the oxygen molecule to form water. During this process, complexes I, III, and IV pump protons to the intermembrane space to generate a proton gradient to drive ATP synthesis by ATP synthase (complex V). During OXPHOS, electrons may also leak and interact with oxygen to form superoxide (i.e., RSO).

**Figure 2 viruses-15-02359-f002:**
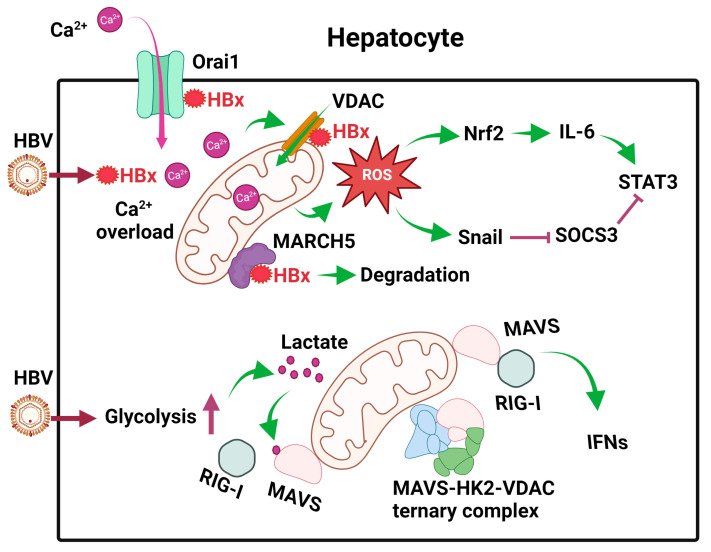
Effects of HBV on mitochondrial metabolism in hepatocytes. After the infection of hepatocytes, HBV expresses its HBx protein, which binds to Orai1 to promote the uptake of Ca^2+^, which then enters mitochondria to cause calcium overload and activate calcium signaling. HBx also binds to VDAC to induce ROS. MARCH5, an E3 ubiquitin ligase, binds to HBx to promote its degradation. HBV also induces glycolysis in hepatocytes to generate lactate, which binds to MAVS on mitochondrial membranes to disrupt RIG-I signaling. In addition, HBV also promotes the formation of the MAVS-HK2-VDAC ternary complex to prevent MAVS from interacting with RIG-I. The ROS produced by mitochondria induces the expression of Nrf2 to induce the expression of IL-6 and activate STAT3. It also induces the expression of Snail to inhibit the expression of SOCS3, an inhibitor of STAT3. These effects of ROS lead to constitutive activation of STAT3 and cell proliferation.

**Figure 3 viruses-15-02359-f003:**
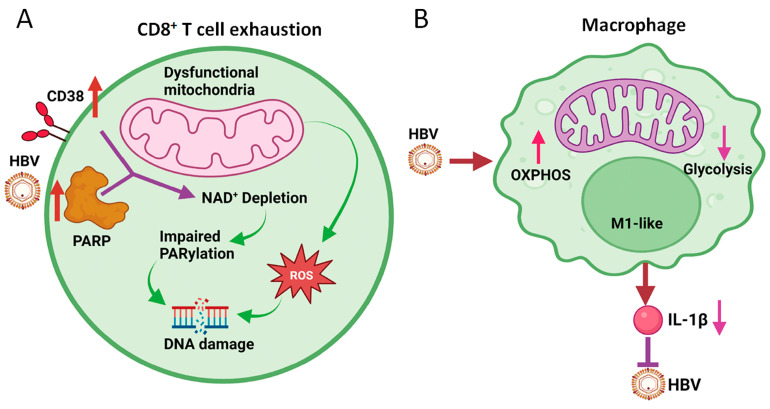
Effects of HBV on mitochondrial metabolism of CD8+ T cells and macrophages. (**A**) HBV induces the expression of CD38 and PARP1, resulting in the depletion of NAD^+^ and the subsequent inhibition of PARylation to cause DNA damage. ROS generated by dysfunctional mitochondria also cause DNA damage. This results in CD8^+^ T cell exhaustion. (**B**) HBV induces OXPHOS and inhibits glycolysis in M1-like macrophages to attenuate the production of IL-1β, which inhibits HBV replication.

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
