# Peer review of "Regulation of Mitochondrial Metabolism by Hepatitis B Virus"

_viruses, 2023, doi:10.3390/v15122359_

Round 1
Reviewer 1 Report
Comments and Suggestions for Authors
This is a good summary of the current state on the effects of HBV infection on mitochondria. I have two minor points for the authors to address.
1. The second paragraph of Section 2. Only about 1% Dane particles contain the DNA genome and the rest are empty. This needs to be made more clear.
2. Section 3.2 and 3.3. Since HBV does not infection CD8 T cells or macrophages, it would be good to have some discussion on how HBV may affect T cells and macrophages.
Reviewer 2 Report
Comments and Suggestions for Authors
General Comments
Li and Ou provide a brief, but comprehensive summary of the interplay between HBV and mitochondrial metabolism. The review briefly summarises the viral life cycle, before breaking down the various viral and host components that regulate mitochondrial function in the context of viral infection. Overall, the review is well written and provides a useful summary for those interested in HBV. I would like to thank the authors for taking the time to go through the literature so thoroughly. Whilst the work is clearly reflecting the considerable body of work that Ou and colleagues have contributed to the field, Li and Ou do well to also consider the other major works that describe the importance of mitochondria in HBV infection.
Given that this review is to be published in a special edition focussed on mitochondria, rather than HBV specifically, it could be said that a little more description of the viral life cycle may be needed. Several descriptions are reliant on some level of background knowledge, which several readers may not have.
The manuscript is not without some typographical errors, and several have been pointed out in the comments below. This is probably not exhaustive, and the review would benefit from a thorough proof read – but this would not be considered as a barrier to publication. There are detailed comments on each section below, of which some should be considered.
Introduction
It would be nice, but by no means essential to include a schematic diagram in the introduction. This could describe the pathways discussed: OXPHOS, TCA, and ROS with their respective metabolic outputs, as well as the various viral perturbations listed afterwards. Whilst not essential it would make for a nice diagram that would accompany Figure 1 to provide some context to the review.
The authors should clarify that the list of viruses with their associated mitochondrial perturbations is not exhaustive. It could also be stated that viruses have been reported to perturb host cell metabolism by other pathways, such as modulation of lipids via Apo family members in HCV infection, as an example.
HBV and its life cycle
Line 46: It is my understanding that the title should be written as ‘life cycle’ as opposed to ‘lifecycle’.
Line 52: the use of ‘the’ before vertical and horizontal transmission can be omitted.
Line 66: state relaxed circular DNA (rcDNA).
Line 67: When describing the genome structure of HBV, it would be convention to list the gene products in size order, C, P, S and X.
Line 72-3: The authors should consider merging these 2 short sentences to something like ‘In contrast to the infectious Dane particles, the 22mm are largely non-infectious empty virions, and other viral particles….’
Line 75: revise to a single sentence ‘HBV is a hepatotropic virus; entering into hepatocytes…’
Line 79: The authors could take the opportunity to state here that integrated forms of the genome are also major sources of viral RNAs, despite being replication deficient. Work from Suslov et al, Freitas et al or Magri et al could be stated here.
Lines 81-84: The authors refer the pol as a DNA polymerase, whereas previously they call it the RT. It would help readability to a wider audience if this was kept consistent.
HBV and Host Metabolism
The description of active and inactive HBV replication in the opening sentences is not overly helpful – perhaps the authors should briefly provide some context on disease status, and instead refer to HBeAg status as positive and negative. This would be more consistent with the EASL guidelines. This is also the case in line 96, where ‘Immune Tolerance phase’ is stated – this is older nomenclature, and not immediately obvious to a reader who is unfamiliar with HBV clinical presentations.
Effect of HBV on the metabolism of hepatocytes
This section is very nicely written, thank you.
The authors should comment on the caveats here: delivery of an overlength genome via adenoviral vector obtains a good transduction efficiency, but is not the most physiological system. Similarly, rats are not natural hosts for HBV, so this would limit their applicability.
Role of HBx on mitochondria metabolism in hepatocytes
The introduction to the role of HBx is incredibly brief, given the vast literature on host processes where HBx has been implicated, the authors could expand a little and provide the reader with more background on the reported roles of X in cell cycle, epigenetic regulation, and HCC genesis to name but a few.
Line 122: reads tha as opposed to that.
Line 126: Given the authors state that depolarisation is regulated by NFKB status, they should comment on the impact of HBx on NFKB.
A nicely worded section, and given the concluding paragraph stating other roles of HBx, it reinforces the request for additional description of reported interplay between HBx and host as an introduction.
Role of mitochondria in innate immune reponse against HBV
Overall, this section is beautifully written and provides a very nice summary on the role of mitochondria and the innate immune response, thank you.
Title should be corrected to ‘the innate immune…’ or ‘innate immune responses’.
Line 139: immun should read immune
Integration of HB DNA into mitochondrial DNA
Line 186: authors should clarify that the non-coding d-loop is a feature of mitoDNA, not a viral genome modification
HBV and mitophagy
Do the authors know whether the relocation of mitochondria has been reported for other viruses? It is often discussed that viruses can modify the host cell to facilitate their infectious life cycle, but the physical reordering of organelles seems like it goes beyond this. One would imagine that highly replicative and cytopathic viruses are capable of clustering mitochondria too. A sentence or two to discuss whether this is HBV specific, or not, would be a worth while addition.
HBV preS-2 mutant and mitochondrial metabolism
Is this in frame N-terminus mutation usually associated with drug resistance, or is it naturally occurring? If the latter, it could suggest a strategy by which the virus manipulates mitochondrial activity.
Line 225: authors should clarify whether the deleted fragment contains the usual T- and B- cell epitopes, or whether the truncated L-HBs is the source of epitopes.
Effect of HBV on the metabolism of CD8 cells
This is an important section of the review, as T cell exhaustion is a major challenge in clinical disease. Surely this provides a barrier for a kick and kill like strategy, as the T cell metabolism may already render them dysfunctional, irrespective of epitope recognition and presentation. Whilst the mitochondria are clearly implicated, other metabolic pathways may also be perturbed. Mala Maini’s group in the UK has published work on cholesterol modifying agents in T cell activity on HBV. Perhaps metabolic changes in these cells, beyond the pathways regulated by mitochondria, are also important in T cell fatigue, this could be discussed here.
Effect of HBV on the metabolism of macrophages
The authors need to check the M1 and M2 polarization description in lines 281-2. They state that exposure to maternal HBV antigens causes the polarization to both M1 and M2. One assumes that one of these should be a negative statement? A minor revision is definitely required here.
Line 296: the authors cite their own work here, which is absolutely fine, however they should state which model they used. Was it an in vitro or an in vivo model system.
Reviewer 3 Report
Comments and Suggestions for Authors
In this manuscript, the authors summarized current research progresses on the interaction between HBV and mitochondrial metabolism, and the effect of this interaction on HBV replication and persistence. They discussed the roles of HBx and preS2 mutant in HBV-mediated mitochondrial metabolism dysfunction in hepatocytes, as well as how HBV modulates mitochondrial metabolism to evade immune response. They also discussed the integration of HBV DNA into mitochondrial DNA and HBV-mediated mitophagy. They claimed that HBV impairs mitochondrial functions to promote HBV replication, persistence and pathogenesis. They further evaluated the effects of HBV on mitochondrial metabolism of CD8+ T cells and macrophages, emphasizing that HBV disrupts the antiviral activities of host immune cells.
Overall, this paper is well written, providing detailed information on the pathways utilized by HBV to affect mitochondrial metabolism both in hepatocytes and immune cells. Therefore, I believe this manuscript meets the requirements for publication in the Viruses.
Minor points
There are some typos in the manuscript.
1. Page 3, Line 146, “In this elgant study” should be “In this elegant study”.
2. Page 5, Line 195, “it was suggested that the HBV DNA integration migth be mediated by HBV” should be “it was suggested that the HBV DNA integration might be mediated by HBV”.
3. Page 5, Line 221, “HBV preS-2 mutant and mitochondrial metabolism” should be “HBV preS2 mutant and mitochondrial metabolism”.
Comments on the Quality of English LanguageThere are several minor typos.
